# ESCRT-III activation by parallel action of ESCRT-I/II and ESCRT-0/Bro1 during MVB biogenesis

Shaogeng Tang[1,2], Nicholas J Buchkovich[1,2†], W Mike Henne[1,2‡], Sudeep Banjade[1,2], Yun Jung Kim[1,2], Scott D Emr[1,2*]

[1]Weill Institute for Cell and Molecular Biology, Cornell University, Ithaca, United States; [2]Department of Molecular Biology and Genetics, Cornell University, Ithaca, United States

*For correspondence: sde26@cornell.edu

Present address: †Department of Microbiology and Immunology, The Pennsylvania State University College of Medicine, Hershey, United States; ‡Department of Cell Biology, The University of Texas Southwestern Medical Center, Dallas, United States

Competing interests: The authors declare that no competing interests exist.

**Abstract** The endosomal sorting complexes required for transport (ESCRT) pathway facilitates multiple fundamental membrane remodeling events. Previously, we determined X-ray crystal structures of ESCRT-III subunit Snf7, the yeast CHMP4 ortholog, in its active and polymeric state (*Tang et al., 2015*). However, how ESCRT-III activation is coordinated by the upstream ESCRT components at endosomes remains unclear. Here, we provide a molecular explanation for the functional divergence of structurally similar ESCRT-III subunits. We characterize novel mutations in ESCRT-III Snf7 that trigger activation, and identify a novel role of Bro1, the yeast ALIX ortholog, in Snf7 assembly. We show that upstream ESCRTs regulate Snf7 activation at both its N-terminal core domain and the C-terminus α6 helix through two parallel ubiquitin-dependent pathways: the ESCRT-I-ESCRT-II-Vps20 pathway and the ESCRT-0-Bro1 pathway. We therefore provide an enhanced understanding for the activation of the spatially unique ESCRT-III-mediated membrane remodeling.

## Introduction

The endosomal sorting complex required for transport (ESCRT) pathway mediates topologically unique membrane budding events. In multivesicular body (MVB) biogenesis, ESCRT-0, I and II sort ubiquitinated cargo by binding ubiquitin and endosomal lipids. ESCRT-III assembles into spiraling polymers for cargo sequestration, and together with the AAA-ATPase Vps4, remodels the membranes to generate cargo-laden intralumenal vesicles (ILVs).

ESCRT-III is a metastable and conformationally dynamic hetero-polymer of four 'core' subunits, Vps20, Snf7/Vps32, Vps24 and Vps2 (*Babst et al., 2002*). All subunits share a common domain organization of an N-terminal helical core domain and a flexible C-terminus, but provide distinct functions. ESCRT-II engages Vps20 to nucleate the polymerization of the most abundant ESCRT-III subunit, Snf7, which then recruits Vps24 and Vps2 (*Teis et al., 2008*). Finally, Vps2 engages Vps4 for ESCRT-III disassembly (*Obita et al., 2007*).

How is Snf7 activated to promote ESCRT-III assembly and cargo sequestration? Previous studies have shown that ESCRT-II and Vps20 modulate Snf7 protofilaments, emphasizing a role of the upstream ESCRTs in defining the assembly and architecture of the ESCRT-III complex (*Henne et al., 2012*; *Teis et al., 2010*). Recently, we have determined X-ray crystal structures of Snf7 protofilaments in the active conformation (*Tang et al., 2015*). Here, using genetics and biochemistry, we identify two parallel ubiquitin-dependent pathways that regulate Snf7 activation through both the Snf7 N-terminal core domain and the C-terminal α6 helix, providing an enhanced understanding of the activation of ESCRT-III-mediated membrane remodeling at endosomes.

## Results

### The α1/2 hairpin confers Vps20 with a unique identity

Although Vps20 and Snf7 display a high degree of homology, they cannot complement each other. In order to identify regions of Vps20 essential for its function, we designed a series of Vps20-Snf7 chimeras and analyzed them by an established quantitative Mup1-pHluorin MVB sorting assay (*Henne et al., 2012*). Although a full-length Vps20 is required for function, retaining only the α1/2 hairpin of Vps20 while replacing the remainder of Vps20 with Snf7 (Vps20$^{1-105}$-Snf7$^{107-240}$) is sufficient for sorting, albeit at ~70% efficiency (*Figure 1A*, *Figure 1—figure supplements 1–2*), suggesting that α1/2 is the minimal region unique to Vps20. This is consistent with the role of α1 of Vps20 in binding to the ESCRT-II subunit Vps25 (*Im et al., 2009*).

### Screening for Vps20-independent Snf7 activation mutants

To investigate the role of Vps20 in nucleating Snf7 *in vivo*, we next applied an unbiased random mutagenic approach. We performed error-prone polymerase chain reaction on *SNF7* and selected mutants that suppress the *vps20Δ* phenotype by growth on L-canavanine (*Figure 1B*). Two *snf7* point mutations in conserved residues, *snf7$^{Q90L}$* and *snf7$^{N100I}$*, showed a partial rescue of the canavanine sensitivity of *vps20Δ* (*Figures 1C-1D*). Remarkably, in 'closed' Snf7, Gln90 of α2 is proximal to α4 (*Tang et al., 2015*), and Asn100 is an asparagine cap of the α2 helix (*Figure 1E*). We propose that these mutations destabilize closed Snf7 by displacing α4 from α2 and extending the α2/3 helix.

Since conformationally active Snf7 resides on membranes, we performed liposome sedimentation assays. As predicted, Q90L enhances Snf7 membrane association from 41% to 78% (*Figure 1F*). To further identify whether these substitutions trigger 'opening' in the core domain, we applied circular dichroism (CD) spectroscopy (*Greenfield, 2006*; *Peter et al., 2004*) on Snf7$^{α1-α4}$, a truncated Snf7 construct with reduced membrane binding compared to the full-length proteins (*Buchkovich et al., 2013*). In the presence of liposomes, we observed a decrease of the negative absorption band at 208 nm and an increase at 222 nm in Q90L and N100I mutants, indicating an increase of α-helicity (*Figure 1G*). These data agree with the hypothesis that Snf7$^{Q90L}$ and Snf7$^{N100I}$ trigger structural rearrangements, where the α2/3 loop becomes α-helical and extends into one elongated α-helix (*Figure 1—figure supplement 3*) as observed in the open structures (*McCullough et al., 2015*; *Tang et al., 2015*). Notably, this structural rearrangement still occurs only upon membrane binding. Moreover, *snf7$^{Q90L}$* and *snf7$^{N100I}$* complement *snf7Δ in vivo*, and Snf7$^{Q90L}$ assembles into protofilaments *in vitro* (*Figure 1—figure supplement 4*), confirming a functional role of the mutants in activating Snf7.

### Auto-activated Snf7 bypasses Vps20

Given that *snf7$^{Q90L}$* and *snf7$^{N100I}$* only modestly suppress *vps20Δ*, we hypothesized that a more stabilized 'open' Snf7 on endosomal membranes would improve the suppression. We combined the activation mutations with R52E (*Henne et al., 2012*) to further trigger 'opening', and swapped α0 of Snf7 with the N-terminal myristoylation motif of Vps20 to enhance its membrane-binding affinity (*Buchkovich et al., 2013*). This yielded *myr-snf7$^{R52E\ Q90L}$* and *myr-snf7$^{R52E\ Q90L\ N100I}$*, hereafter denoted as *snf7\*\** and *snf7\*\*\**, which sorted cargo with increased efficiencies, albeit not completely restoring wild-type levels (*Figure 2A*, *Figure 2—figure supplements 1–2*).

Consistent with these observations, the ESCRT-dependent cargo GFP-Cps1 partially localized to the vacuolar lumen in *vps20Δ* with *snf7\*\** or *snf7\*\*\** (*Figure 2B*), indicating a substantial level of MVB sorting. Moreover, *snf7\*\** and *snf7\*\*\** were also able to rescue the canavanine sensitivity of *vps20Δ* (*Figure 2C*). Thus, these *snf7* suppressors exhibit the ability to sort cargo at MVB.

To visualize whether the *snf7* suppressors could produce ILVs *in vivo*, we utilized a temperature sensitive allele of the vacuolar SNARE *vam7* to accumulate MVBs and examined yeast with thin-section TEM (*Buchkovich et al., 2013*; *Sato et al., 1998*) (*Figure 2D*). We observed that while ILVs in wild-type cells have a diameter of ~32 nm, *snf7\*\** and *snf7\*\*\** show a decrease in ILV number and an increase in ILV diameter to ~43 nm (*Figures 2E–F*, See Materials and methods). Since ESCRT-II and Vps20 set the architecture of ESCRT-III, we propose that the variation in ILV size is a result of aberrant ESCRT-III architecture, although we cannot completely rule out the possibility of changes in dynamics of ESCRT-III disassembly by Vps4 (*Nickerson et al., 2010*).

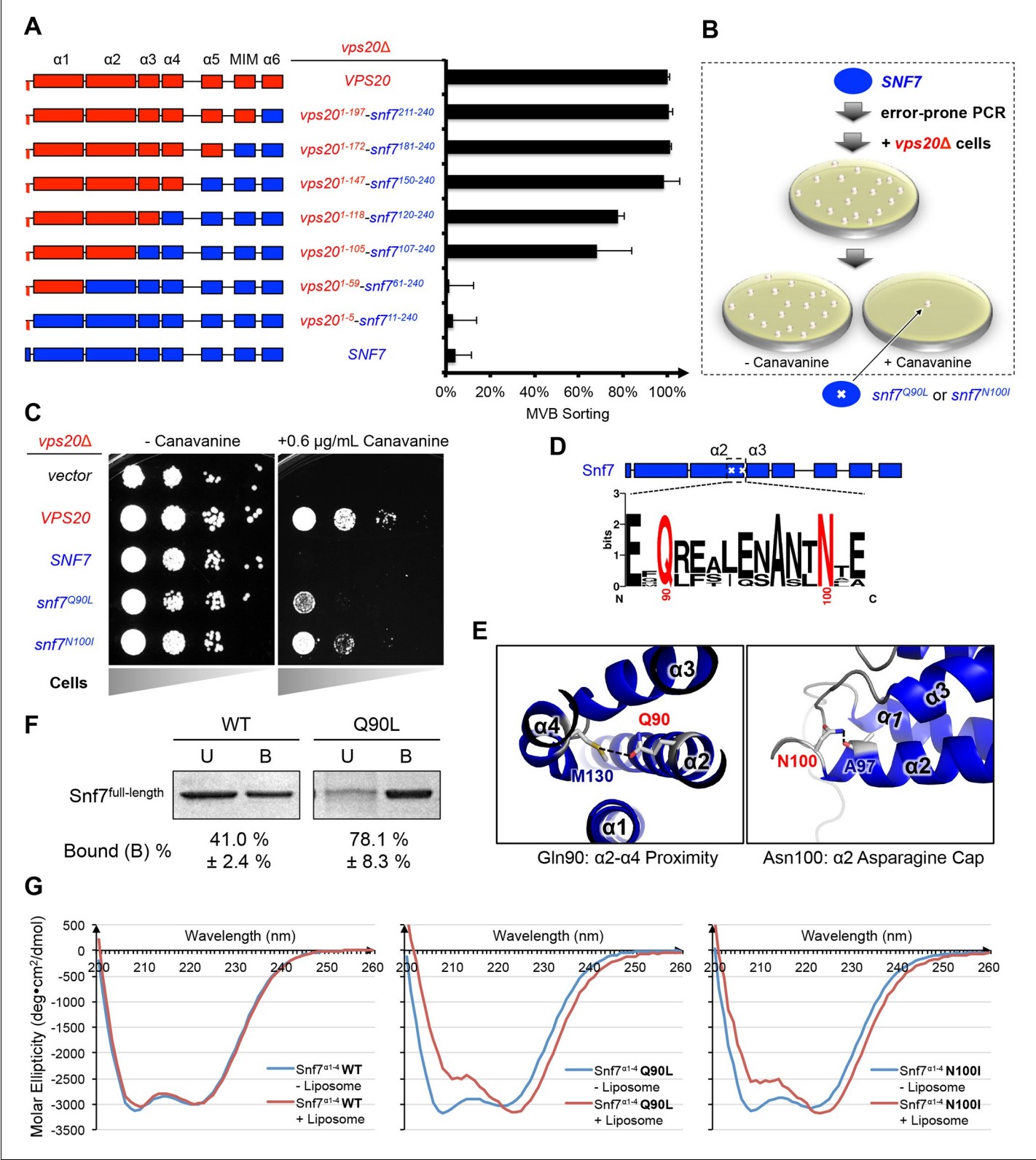

**Figure 1.** Novel Snf7 point mutations trigger core domain activation. (**A**) Domain organization of Vps20(red)-Snf7(blue) chimera (left) and quantitative MVB sorting data (right) for *vps20Δ* yeast exogenously expressing *VPS20*, *vps20¹⁻¹⁹⁷-snf7²¹¹⁻²⁴⁰*, *vps20¹⁻¹⁷²-snf7¹⁸¹⁻²⁴⁰*, *vps20¹⁻¹⁴⁷-snf7¹⁵⁰⁻²⁴⁰*, *vps20¹⁻¹¹⁸-snf7¹²⁰⁻²⁴⁰*, *vps20¹⁻¹⁰⁵-snf7¹⁰⁷⁻²⁴⁰*, *vps20¹⁻⁵⁹-snf7⁶¹⁻²⁴⁰*, *vps20¹⁻⁵-snf7¹¹⁻²⁴⁰*, and *SNF7*. Error bars represent standard deviations from 3–5 independent experiments. (**B**) Screening strategy to identify *snf7* suppressors in *vps20Δ* yeast. (**C**) Canavanine sensitivity assay for *vps20Δ* yeast exogenously

*Figure 1 continued on next page*

*Figure 1 continued*

expressing empty vector, *VPS20*, *SNF7*, *snf7*$^{Q90L}$, and *snf7*$^{N100I}$. (**D**) Domain organization of Snf7, with the locations of Gln90 and Asn100. WebLogo of protein sequence analysis (**Doerks et al., 2002**) of Snf7 orthologs from *Saccharomyces cerevisiae, Homo sapiens, Mus musculus, Xenopus laevis, Drosophila melanogaster, Caenorhabditis elegans, Schizosaccharomyces pombe*. (**E**) Close-up view of the side chain interactions of Gln90 (left) and Asn100 (right) in a 'closed' Snf7 homology model (**Henne et al., 2012**). (**F**) Liposome sedimentation assays of Snf7$^{WT}$ and Snf7$^{Q90L}$. Liposome-bound (**B**) proteins and unbound (**U**) proteins. (**G**) CD scanning spectra from 200 nm to 260 nm of wild-type Snf7$^{\alpha 1-4}$ (left), Snf7$^{\alpha 1-4}$ Q90L (middle), and Snf7$^{\alpha 1-4}$ N100I proteins with (red) and without (blue) liposomes.

The following figure supplements are available for figure 1:

**Figure supplement 1.** A full-length Vps20 is required for MVB sorting.

**Figure supplement 2.** Vps20-Snf7 chimera complements Vps20 function.

**Figure supplement 3.** Snf7$^{Q90L}$ and Snf7$^{N100I}$ trigger core domain activation.

**Figure supplement 4.** Snf7$^{Q90L}$ assembles into protofilaments *in vivo* and *in vitro*.

## Auto-activated Snf7 bypasses ESCRT-I and ESCRT-II

Intrigued by the *vps20∆* suppression, we next wanted to test if these auto-activated Snf7 mutants could also bypass the loss of other ESCRT components (*Figure 3A*). Among them, the downstream ESCRT-III subunits Vps24 and Vps2 are known to modulate Snf7 architecture (**Henne et al., 2012**; **Teis et al., 2008**) and recruit the AAA-ATPase Vps4 via their C-terminal MIM motifs for ESCRT-III disassembly (**Obita et al., 2007**). We found that auto-activated Snf7 does not suppress *vps24∆*, *vps2∆* or *vps4∆* (*Figure 3B*, *Figure 3—figure supplement 1*). This is consistent with the role of the suppressors in activating but not modulating or disassembling Snf7 filaments, reinforcing the division of labor among ESCRT-III subunits.

Previous studies showed that ESCRT-III assembly is regulated by ESCRT-II (**Henne et al., 2012**; **Teis et al., 2010**) (*Figure 3A*). ESCRT-II is a Y-shaped heterotetramer of Vps36, Vps22 and two Vps25 (arms). Vps36 GLUE domain binds ubiquitinated cargo and endosome-specific phosphatidylinositol 3-phosphate, PI3P; and each Vps25 'arm' binds one molecule of the ESCRT-III nucleator, Vps20. Since repurposing Snf7 to bind ESCRT-II does not improve the suppression (*Figure 3—figure supplement 2*), we next tested the functionality of the suppressors in ESCRT-II single and double deletion mutants. Strikingly, *snf7\*\** and *snf7\*\*\** resulted in better suppression in ESCRT-II deletion compared to *vps20∆*, with sorting efficiencies of ~60%–70% (*Figure 3C*, *Figure 3—figure supplements 3–4*).

We next tested ESCRT-I mutants. ESCRT-I is a heterotetramer of Vps23, Vps28, Vps37 and Mvb12. Vps23 UEV domain recognizes ubiquitinated cargo, Vps37 N-terminal helix binds to membranes, and Vps28 CTD engages Vps36 GLUE domain of ESCRT-II. We expressed the suppressors in ESCRT-I single and ESCRT-I/II double deletion mutants and we observed near wild-type sorting efficiencies (*Figure 3D*) with enlarged ILV sizes (*Figure 3—figure supplements 5–6*). Our data suggest that ESCRT-I and ESCRT-II set up the ESCRT-III architecture to program vesicle dimension.

## Auto-activated Snf7 does not bypass Bro1 and ESCRT-0

Because ESCRT-I and ESCRT-II cluster ubiquitinated cargo prior to their packaging into ILVs, the observed suppression indicated that the auto-activated Snf7 might sort cargo in a ubiquitin-independent manner. We next tested whether auto-activated Snf7 could bypass the remaining ubiquitin-binding ESCRT components, ESCRT-0 (Vps27 and Hse1) and, the yeast ALIX ortholog, Bro1/Vps31. Interestingly, the engineered *snf7* suppressors do not sort cargo in *vps27∆* or *bro1∆* (*Figures 3E and G*), or *hse1∆* in combination with *vps20∆*, *vps25∆* (ESCRT-II) or *vps23∆* (ESCRT-I) (*Figure 3F*). To test whether ubiquitin-binding of ESCRT-0 and Bro1 is critical, we expressed ubiquitin-binding mutants *vps27*$^{S270D\ S313D}$ (*vps27*$^{UIM}$) and *bro1*$^{I377R\ L386R}$ (*bro1*$^{UBD}$) (**Bilodeau et al., 2002**; **Pashkova et al., 2013**). They reduced the functionality of *snf7\*\*\** in *vps20∆*, *vps25∆* or *vps23∆* (*Figure 4A*, *Figure 4—figure supplement 1*). These data suggest that despite the ESCRT-I/II-independence, the suppression is still ubiquitin-dependent (*Figure 3—figure supplement 7*), perhaps

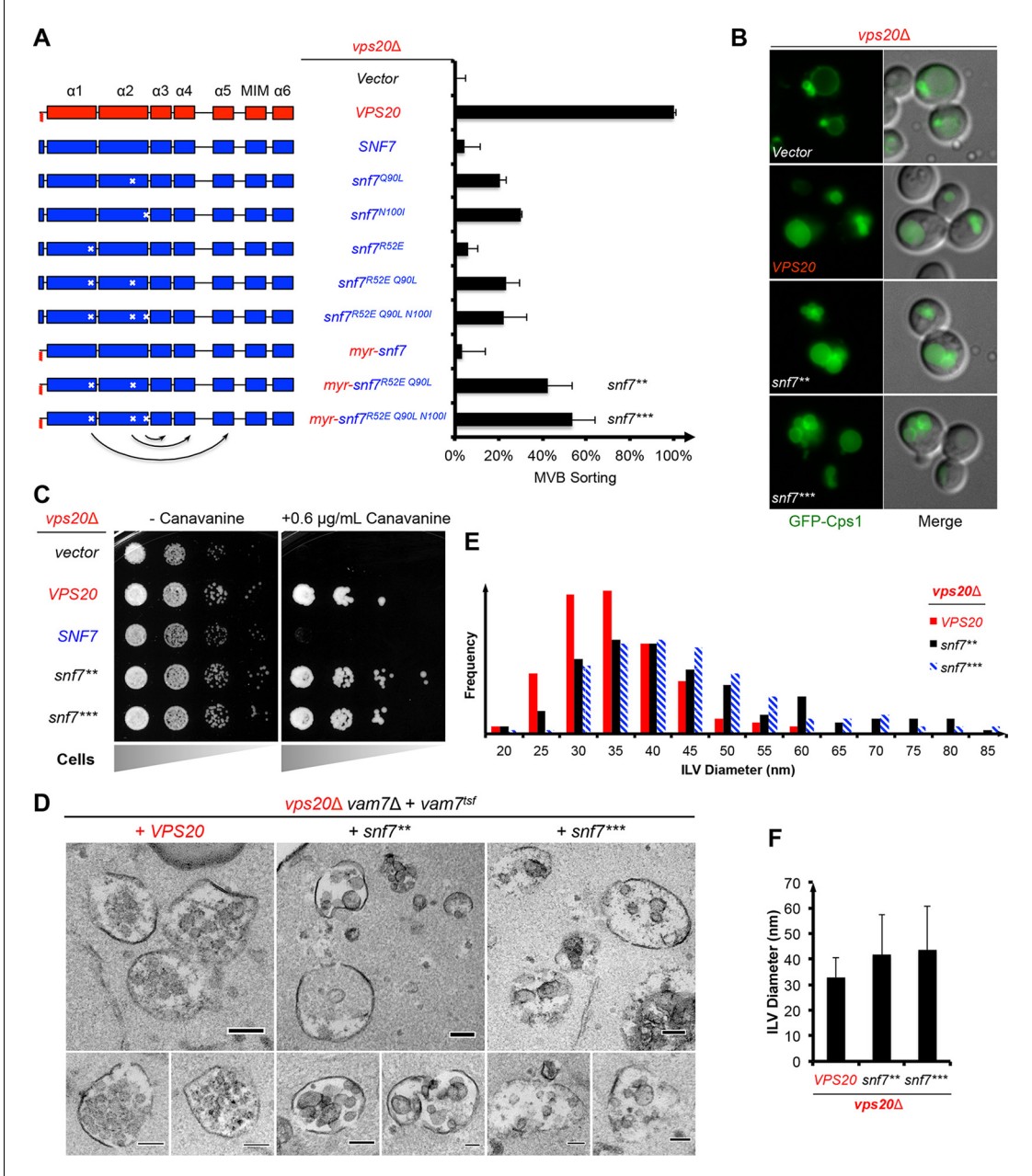

**Figure 2.** Auto-activated Snf7 functionally bypasses Vps20. (**A**) Domain organization of Snf7 mutants (left) and quantitative MVB sorting data (right) for *vps20Δ* yeast exogenously expressing empty vector, *VPS20*, *SNF7*, *snf7$^{Q90L}$*, *snf7$^{N100I}$*, *snf7$^{R52E}$*, *snf7$^{R52E\ Q90L}$*, *snf7$^{R52E\ Q90L\ N100I}$*, *myr-snf7*, *myr-snf7$^{R52E\ Q90L}$*, and *myr-snf7$^{R52E\ Q90L\ N100I}$*. Error bars represent standard deviations from 3–5 independent experiments. The data of *myr-snf7* (*vps20$^{1-5}$-snf7$^{11-240}$*) and *SNF7* were re-plotted from **Figure 1A** for comparsion. Mutants *myr-snf7$^{R52E\ Q90L}$* and *myr-snf7$^{R52E\ Q90L\ N100I}$* are referred to *snf7\*\** and *snf7\*\*\**, respectively. (**B**) Representative images of *vps20Δ* yeast exogenously expressing *GFP-CPS1* with *VPS20*, *snf7\*\**, and *snf7\*\*\**. GFP images (left) and composite images of GFP and DIC (right). (**C**) Canavanine sensitivity assay for *vps20Δ* yeast exogenously expressing empty vector, *VPS20*, *SNF7*, *snf7$^{\*\*}$*, and *snf7$^{\*\*\*}$*. (**D**) Representative TEM images of ILV-containing MVBs from *vps20Δ vam7Δ* yeast exogenously expressing *vam7$^{tsf}$*, with *VPS20*, *snf7$^{\*\*}$*, and *snf7$^{\*\*\*}$*. Scale bars 100 nm. (**E–F**) Quantitation of ILV (*N*=150 ILV summed per sample) outer diameter from (**D**) in frequency distributions (**E**), and averaged measurements (**F**). Error bars represent standard deviations.

The following figure supplements are available for figure 2:

**Figure supplement 1.** Activation mutants complement *snf7Δ in vivo*.

**Figure supplement 2.** Activation mutants complement *vps20Δ snf7Δ in vivo*.

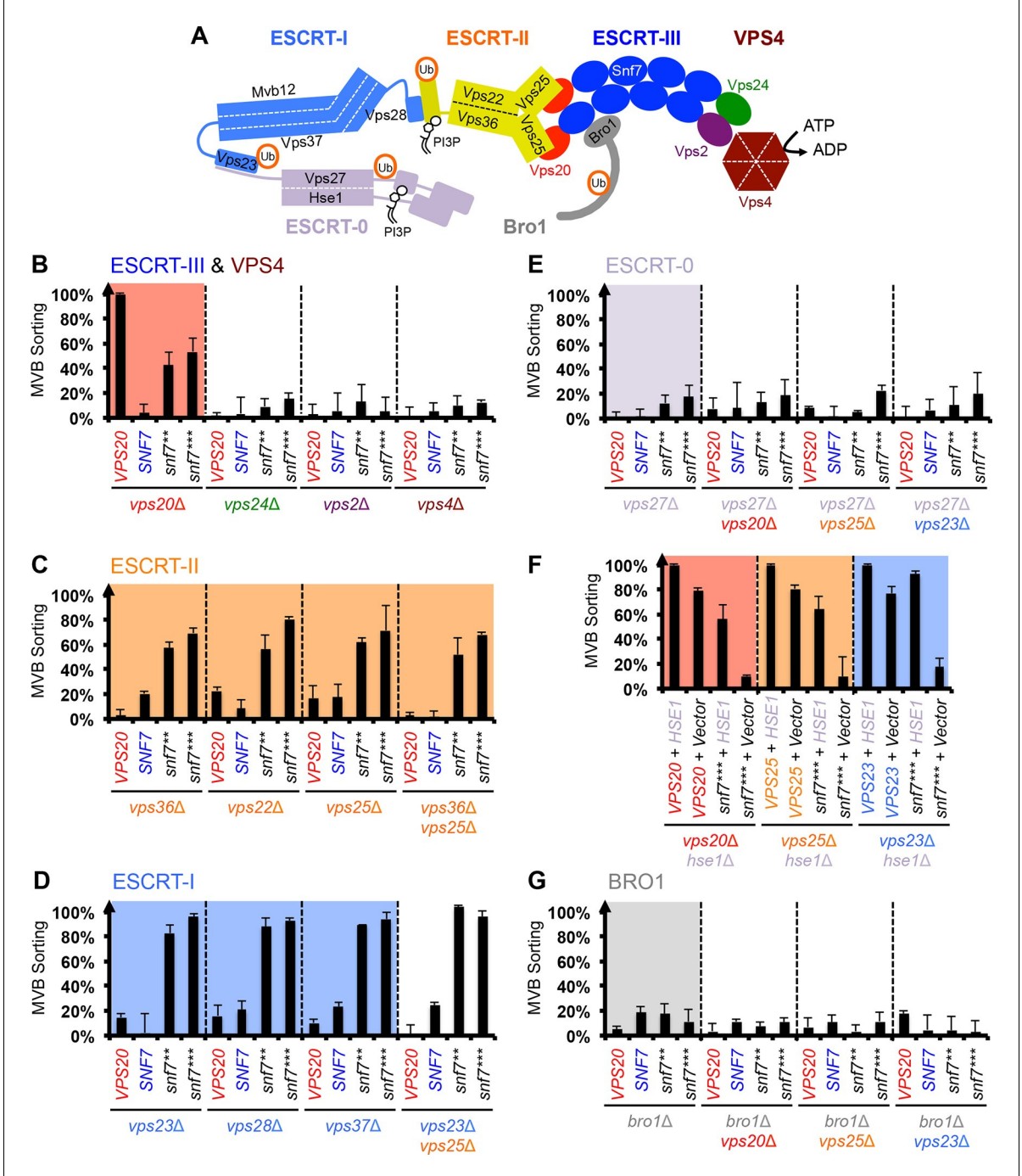

**Figure 3.** Snf7 core domain auto-activation bypasses ESCRT-I and ESCRT-II. (**A**) Cartoon of the ESCRT pathway in MVB Biogenesis. (**B–D** & **F–G**) Quantitative MVB sorting data for (**B**) *vps20Δ*, *vps24Δ*, *vps2Δ* and *vps4Δ* yeast, (**C**) *vps36Δ*, *vps22Δ*, *vps25Δ* and *vps36Δ vps25Δ* yeast, (**D**) *vps23Δ*, *vps28Δ*, *vps37Δ* and *vps23Δ vps25Δ* yeast, (**E**) *vps27Δ*, *vps27Δ vps20Δ*, *vps27Δ vps25Δ* and *vps27Δ vps23Δ* yeast, (**G**) *bro1Δ*, *bro1Δ vps20Δ*, *bro1Δ vps25Δ* and *bro1Δ vps23Δ* yeast exogenously expressing *VPS20, SNF7, snf7\*\**, and *snf7\*\*\**, respectively. The data from *vps20Δ* were partially re-plotted from *Figure 2A* for comparison. (**F**) Quantitative MVB sorting data for *vps20Δ hse1Δ*, *vps25Δ hse1Δ*, and *vps23Δ hse1Δ* yeast exogenously expressing *VPS20/VPS25/VPS23* and *HSE1*, and *VPS20/VPS25/VPS23* and empty vector, *snf7\*\*\** and *HSE1*, and *snf7\*\*\** and empty vector, respectively. Error bars represent standard deviations from 3–5 independent experiments.

The following figure supplements are available for figure 3:

**Figure supplement 1.** Snf7 core domain auto-activation does not suppress *vps24Δ*, *vps2Δ* and *vps4Δ*.

*Figure 3 continued on next page*

*Figure 3 continued*

**Figure supplement 2.** Repurposing Snf7 to bind Vps25 does not improve suppression in *vps20Δ*.
**Figure supplement 3.** Snf7 core domain auto-activation suppresses ESCRT-II deletions.
**Figure supplement 4.** Snf7 core domain auto-activation suppresses ESCRT-II 'arm' mutants.
**Figure supplement 5.** Snf7 core domain auto-activation suppresses ESCRT-I deletions.
**Figure supplement 6.** MVB morphologies of *snf7\*\** and *snf7\*\*\** in *vps23Δ*.
**Figure supplement 7.** Localization of Vph1-GFP and GFP-Cps1[K8R K12R].

through another subset of machinery of ESCRT-0 and Bro1. We thus propose that ESCRT-0/Bro1 are required to sort ubiquitinated cargo for ESCRT-III sequestration in parallel to ESCRT-I/II.

## Bro1 binds to Snf7 α6 helix and activates Snf7

Bro1 has been shown to directly interact with Snf7, and X-ray crystal structures suggest that the C-terminal α6 helix of Snf7 binds to the Bro1 domain of Bro1 (*Kim et al., 2005*; *McCullough et al., 2008*; *Wemmer et al., 2011*). To test whether this interaction is required for *snf7* suppression, we mutated residues at the Snf7-Bro1 interface. Notably, neither the Bro1-binding defective Snf7\*\*\* [L231K L234K] mutant (*snf7\*\*\* [BRO1]*), nor the Snf7-binding defective Bro1[I144D L336D] mutant (*bro1[SNF7]*), suppresses *vps20Δ*, *vps25Δ* or *vps23Δ* (*Figure 4B*, *Figure 4—figure supplement 1*). This strongly suggests that α6 of Snf7 is also auto-inhibitory, and that a physical binding between Snf7 α6 and Bro1 is a prerequisite for Snf7 activation.

We next tested if the Snf7-Bro1 interaction would release the α6 auto-inhibition. While the recombinant Snf7[WT] does not assemble due to auto-inhibition, coincubation with Bro1 resulted in Snf7 protofilament assembly (*Figure 4C*), indicating that Bro1 directly triggers Snf7 activation. In agreement with this, the α6 truncated Snf7 (Snf7[1-225]) releases auto-inhibition and assembles into protofilaments (*Figure 4D*). Therefore, our data suggest that while Snf7 N100I, Q90L, and R52E release auto-inhibition in α3, α4, and α5, respectively, α6 of Snf7 is also auto-inhibitory and its activation is Bro1-dependent (*Figure 4E*).

## Discussion

The ancient and conserved ESCRT-III membrane-remodeling machinery plays a critical role in numerous fundamental cellular processes, including MVB biogenesis, viral budding and cytokinesis. Building on our previous study (*Tang et al., 2015*), we focused on the predominant ESCRT-III subunit, Snf7, to understand the molecular mechanisms governing ESCRT-III for its dynamic conversion from an auto-inhibited soluble monomer to a membrane-bending polymer. Remarkably, a recent cryo-EM study on ESCRT-III IST1/CHMP1B co-polymer suggested that CHMP1B (Did2/Vps46) undergoes a similar structural rearrangement for assembly (*McCullough et al., 2015*), implying that the core domain extension is a common theme of ESCRT-III activation.

Here, using a mutagenic approach, we identified novel Snf7 point mutations that release the auto-inhibition of α3 and α4 as observed in the conformationally open structures. Surprisingly, this leads to Snf7 activation that functionally bypasses the ESCRT-III nucleator Vps20, as well as the ESCRT-II and ESCRT-I complexes. This suggests that Snf7, along with its downstream ESCRT components, Vps24, Vps2 and Vps4, but not ESCRT-I/II, are among the minimal machinery required for membrane remodeling.

Our data suggest that ESCRT-III activation is mediated by two parallel pathways, ESCRT-I/II and ESCRT-0/Bro1 (*Figure 4F*). Bro1, directly triggers ESCRT-III assembly by binding to the C-terminal α6 of Snf7 (*Figure 4C*). Given that ESCRT-0 directly engages Bro1 (*Lee et al., 2016*) to recognize ubiquitinated cargo (*Pashkova et al., 2013*), we showed that Snf7 α6 binding to Bro1 relieves

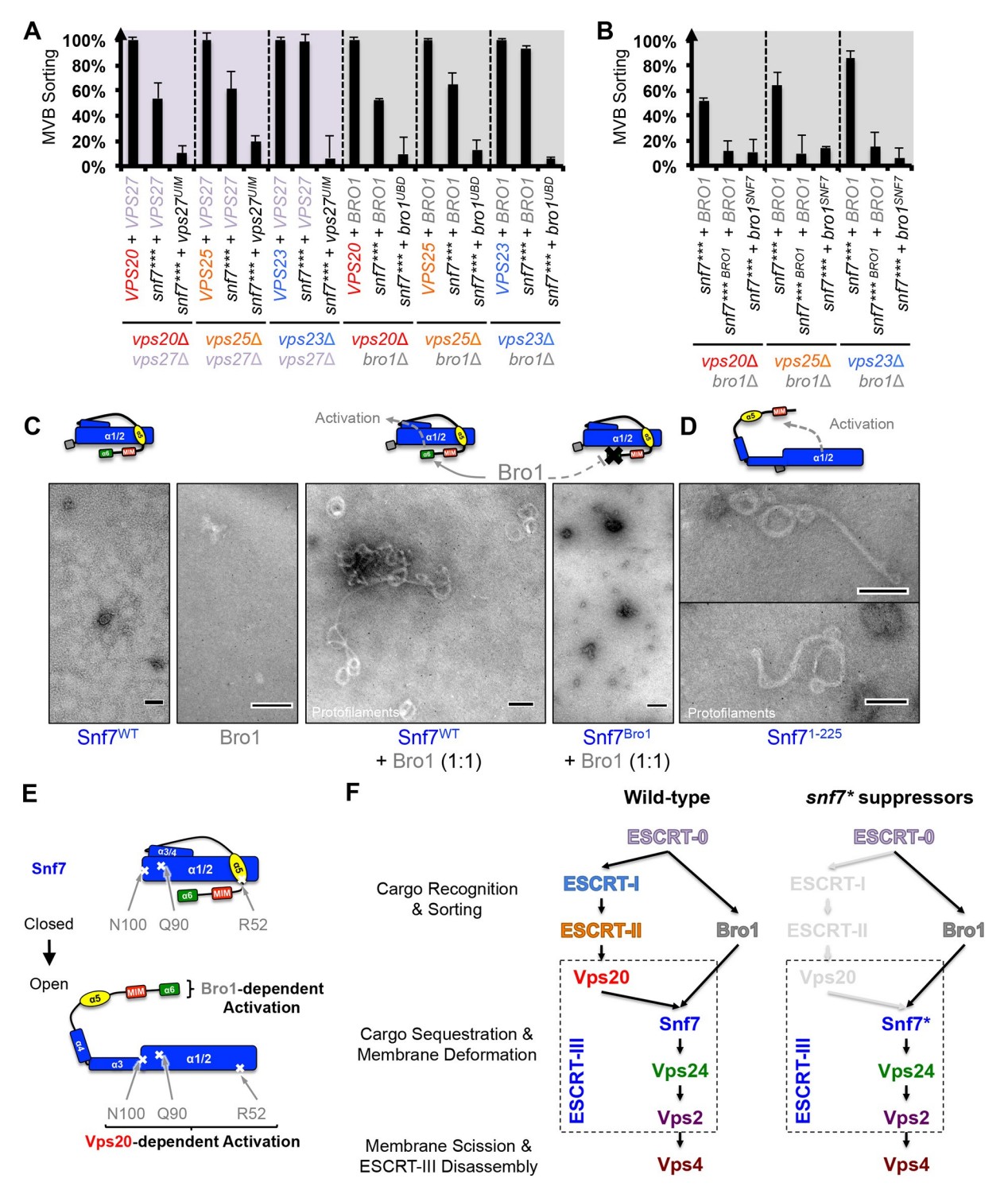

**Figure 4.** Parallel Snf7 activation at its core domain and the C-terminal α6. (A–B) Quantitative MVB sorting data for (A) *vps20Δ vps27Δ*, *vps25Δ vps27Δ*, and *vps23Δ vps27Δ* yeast exogenously expressing *VPS20/VPS25/VPS23* and *VPS27*, *snf7\*\*\** and *VPS27*, and *snf7\*\*\** and *vps27^S270D S313D*(*vps27^UIM*), and *vps20Δ bro1Δ*, *vps25Δ bro1Δ*, and *vps23Δ bro1Δ* yeast exogenously expressing *VPS20/VPS25/VPS23* and *BRO1*, *snf7\*\*\** and *BRO1*, and *snf7\*\*\** and *bro1^I377R L386R*(*bro1^UBD*), respectively, and for (B) *vps20Δ bro1Δ*, *vps25Δ bro1*, and *vps23Δ bro1Δ* yeast exogenously expressing *snf7\*\*\** and *BRO1*, *snf7\*\*\* L231K L234K* (*snf7\*\*\* ^BRO1*) and *BRO1*, and *snf7\*\*\**and *bro1^I144D L336D*(*bro1^SNF7*), respectively. Error bars represent standard deviations from 3–5 independent experiments. (C–D) Representative TEM images of (C) Snf7^WT, Bro1, and Snf7^WT with Bro1 (1:1), and (D) Snf7^1-225 and Snf7^R52E. Scale bars 100 nm. Cartoon diagrams of Snf7 activation. (E) Cartoon diagrams of closed and open Snf7, with the locations of Vps20-dependent activation sites,

*Figure 4 continued on next page*

*Figure 4 continued*

Arg52, Gln90, and Asn100, and Bro1-dependent activation region, α6. (F) Conceptual models of parallel ESCRT-III Snf7 activation pathways in MVB biogenesis of wild-type (left) and the core domain auto-activated Snf7 mutant, Snf7* (right).

The following figure supplement is available for figure 4:

**Figure supplement 1.** Snf7-Bro1 interaction is important for MVB sorting.

autoinhibiton of Snf7. This adds to the roles for Bro1, besides its recruitment of the Doa4 deubiquitinase in the MVB pathway (*Luhtala and Odorizzi, 2004*).

Consistent with our observation, a very recent study suggested that ALIX and ESCRT-I/II function as parallel CHMP4B (Snf7 ortholog in human) recruiters in cytokinetic abscission (*Christ et al., 2016*).

While biochemical data suggest that Snf7 can be activated by specific point mutations in the core domain or truncation at the C-terminus *in vitro*, our genetic evidence indicate that the conformational equilibrium of Snf7 is tightly regulated by two pathways *in vivo* to achieve ubiquitin-dependent cargo sorting at endosomes: 1) ESCRT-I/ESCRT-II/Vps20 activates the N-terminal core domain of Snf7; 2) ESCRT-0/Bro1 activates the C-terminal α6 of Snf7 (*Figures 4E–F*). Our results provide novel insights into a two-stage activation pathway for ESCRT-III-mediated membrane remodeling.

## Materials and methods

### Fluorescence microscopy, canavanine plating assay, western blotting, protein purification and liposome sedimentation

Fluorescence microscopy, western blotting and recombinant Snf7 purification for CD, TEM and liposome sedimentation analysis were performed as described (*Buchkovich et al., 2013*; *Henne et al., 2012*; *Tang et al., 2015*), and canavanine plating assay as described (*Lin et al., 2008*).

For Bro1 purification, *Saccharomyces cerevisiae BRO1* was cloned into the pET23d vector (Novagen, Billerica, MA, USA) with an N-terminal His$_6$-tag, induced by 1 mM IPTG at 18°C overnight from BL21 *E. coli* cells, and purified by TALON metal affinity resin (Clontech). Protein-bound TALON resins were washed in 500 mM NaCl, 20 mM HEPES pH 7.4, 20 mM imidazole, and eluted in 150 mM NaCl, 20 mM HEPES pH 7.4, 400 mM imidazole.

### Flow cytometry

The quantitative Mup1-pHluorin ESCRT cargo-sorting flow cytometry assay was performed as described (*Buchkovich et al., 2013*; *Henne et al., 2012*; *Tang et al., 2015*). Briefly, mid-log yeast cell cultures grown with the addition of 20 μg/mL *L*-methionine for 2 hr were resuspended in 1x PBS buffer. Mean green fluorescence (FL1-A channel) of 100,000 events was recorded and gated on a BD Accuri C6 flow cytometer. For single ESCRT mutants, take *Figure 1A* for example: NBY42 (*vps20Δ MUP1-PH*) yeast cells were transformed with 1) pRS416 empty vector, 2) pRS416 *VPS20*, or 3) different mutants, respectively. Gated mean FL1-A values, *F*, of each sample are recorded and sorting scores are calculated as:

$$MVB \; Sorting \; \% = \left(1 - \frac{F_{Mutant} - F_{VPS20}}{F_{empty\,vector} - F_{VPS20}}\right) \times 100\%$$

Sorting scores of 3 to 5 independent experiments are used to calculate standard deviation.

For double ESCRT mutants, take *Figure 3D* panel *vps23Δvps25Δ* for example. STY64 (*vps23Δ vps25Δ MUP1-PH*) yeast cells were co-transformed with 1) pRS415 empty vector and pRS416 empty vector, 2) pRS415 *VPS25* and pRS416 *VPS23*, 3) pRS415 empty vector and pRS416 *VPS20*, 4) pRS415 empty vector and pRS416 *SNF7*, 5) pRS415 empty vector and pRS416 *snf7\*\**, or 6) pRS415 empty vector and pRS416 *snf7\*\*\**, respectively. MVB sorting scores are calculated as:

$$MVB \; Sorting \; \% = \left(1 - \frac{F_{Mutant} - F_{VPS25+VPS23}}{F_{empty\,vector+empty\,vector} - F_{VPS25+VPS23}}\right) \times 100\%$$

Sorting scores of 3 to 5 independent experiments are used to calculate standard deviation.

## Yeast strain and plasmids

See *Supplementary file 1* for a list of plasmids and yeast strains used in this study.

## *SNF7* random mutagenesis for *vps20Δ* suppressor screening

The DNA sequence of *Saccharomyces cerevisiae SNF7* with 500bp of *5'UTR* and 500bp of *3'UTR* was amplified by Taq DNA polymerase with 20 µM $MnCl_2$ and manipulated dNTP (N=A, T, G, or C) concentrations of 250 µM for three dNTPs and 25 µM for the other dNTP. Four individual 50 µL PCR reactions with different dNTP ratios were mixed, purified and transformed in *vps20Δ* yeast, along with a restriction enzyme digested vector of *3'UTR-pRS416-5'UTR*. Yeast cells were plated and grown on YNB-uracil for 3 days at 26°C, and replica plated on YNB-uracil with 4.0 µg/mL of *L*-canavanine. Canavanine-resistant yeast colonies were selected, and gap-repaired pRS416 *snf7* mutant were prepped, amplified and sequenced.

## Circular dichroism spectroscopy

CD experiments were carried out using an Aviv Biomedical CD spectrometer Model 202–01. 10 µM Snf7$^{core}$ mutants were buffer exchanged by Superdex-200 gel filtration (GE Healthcare Life Sciences) to 10 mM sodium phosphate buffer pH 7.5. For solution samples, Snf7$^{core}$ was mixed with an equal volume of buffer. For liposome samples, Snf7$^{core}$ was mixed with an equal volume of 1.0 mg/mL liposomes of 800 nm diameter, with 60% 1,2-dioleoyl-*sn*-glycero-3-phosphocholine (DOPC), 30% 1,2-dioleoyl-*sn*-glycero-3-phospho-*L*-serine (DOPS), 10% phosphatidylinositol 3-phosphate (PI(3)P). The preparation of liposomes was performed as previously described (*Henne et al., 2012*).

The degrees of ellipticity were measured at 4°C and scanned from 260 nm to 200 nm. Molar ellipticity, $\theta$, was then normalized using the following equation and plotted versus wavelength, where $n=142$ is the number of peptide bonds.

$$\theta \left(deg{\cdot}cm^2/dmol\right) = \frac{Ellipticity(mdeg)}{Pathlength(mm){\cdot}[Protein](\mu M){\cdot}n} \times 10^6$$

## Negative stain transmission electron microscopy

Visualization of ESCRT-III assembly using purified recombinant ESCRT components was performed as previously described (*Henne et al., 2012*).

Visualization of MVB in *vam7$^{tsf}$* yeast cells was performed as previously described (*Buchkovich et al., 2013*). Briefly, 30 ODV of mid-log *vam7$^{tsf}$* yeast cells were grown at 38°C for 3 hr, and then fixed with 2.5% (*v/v*) glutaraldehyde for 1 hr and spheroplasted with zymolyase and gluculase before embedding in 2% ultra-low temperature agarose. Cells were incubated in 1% osmium tetroxide/1% potassium ferrocyanide for 30 min, 1% thiocarbohydrazide for 5 min, and 1% osmium tetroxide/1% postassium ferrocyanide for 5 min. After dehydration through an ethanol series, samples were transitioned into 100% propylene oxide and embedded in Spurr's resin. Note that osmotic gradients during fixation or dehydration might account for the MVB morphological defects and the larger mean ILV diameter compared to samples prepared by high-pressure freezing and automated freeze-substitution. However, all yeast cells used in these experiments were treated equally. All TEM was performed on a Morgnani 268 transmission electron microscope (FEI) with an AMT digital camera.

## Acknowledgements

We gratefully thank Sarah T. Griffin, Yi-Chun Yeh, Leonid A. Timashev, Ming Li and Lu Zhu for technical expertise and sharing reagents. We thank Anthony C Gatts VI for TEM, Nattakan Sukomon and Brian R. Crane for CD spectroscopy. We thank Yuxin Mao for critical reading of the manuscript, and J. Christopher Fromme, Peter P. Borbat and William J. Brown for helpful discussion.

## Additional information

### Funding

| Funder | Grant reference number | Author |
|---|---|---|
| Cornell University | Harry and Samuel Mann Outstanding Graduate Student Award | Shaogeng Tang |
| National Institutes of Health | NIGMS Predoctoral Training Grant in Cellular and Molecular Biology, T32GM007273 | Shaogeng Tang |
| American Cancer Society | Postdoctoral Fellowship, PF-12-062-01-DMC | Nicholas J Buchkovich |
| Cornell University | Sam and Nancy Fleming Research Fellowship | W Mike Henne |
| Cornell University | Research Grant, CU3704 | Scott D Emr |

The funders had no role in study design, data collection and interpretation, or the decision to submit the work for publication.

### Author contributions

ST, NJB, Conception and design, Acquisition of data, Analysis and interpretation of data, Drafting or revising the article, Contributed unpublished essential data or reagents; WMH, Conception and design, Acquisition of data, Analysis and interpretation of data, Drafting or revising the article; SB, Analysis and interpretation of data, Drafting or revising the article, Contributed unpublished essential data or reagents; YJK, Acquisition of data, Analysis and interpretation of data, Contributed unpublished essential data or reagents; SDE, Conception and design, Analysis and interpretation of data, Drafting or revising the article

### Author ORCIDs

Shaogeng Tang, http://orcid.org/0000-0002-3904-492X
Scott D Emr, http://orcid.org/0000-0002-5408-6781

## Additional files

### Supplementary files

• Supplementary file 1.

### Major datasets

The following previously published datasets were used:

| Author(s) | Year | Dataset title | Dataset URL | Database, license, and accessibility information |
|---|---|---|---|---|
| Tang S, Henne WM, Borbat PP, Buchkovich NJ, Freed JH, Mao Y, Fromme JC, Emr SD | 2015 | X-ray Crystal Structure of ESCRT-III Snf7 core domain (conformation A) | http://www.rcsb.org/pdb/explore.do?structureId=5FD7 | Publicly available at the RCSB Protein Data Bank (Accession no: 5FD7). |
| Tang S, Henne WM, Borbat PP, Buchkovich NJ, Freed JH, Mao Y, Fromme JC, Emr SD | 2015 | X-ray Crystal Structure of ESCRT-III Snf7 core domain (conformation B) | http://www.rcsb.org/pdb/explore.do?structureId=5FD9 | Publicly available at the RCSB Protein Data Bank (Accession no: 5FD9). |

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
