## [Decision Letter]

Thank you for submitting your article "ESCRT-III activation mutants mediate MVB biogenesis independent of ESCRT-I and ESCRT-II" for consideration by *eLife*. Your article has been favorably evaluated by John Kuriyan (Senior editor) and three reviewers, one of whom, (William Weis) is a member of our Board of Reviewing Editors.

The following individual involved in review of your submission has agreed to reveal their identity: Alexey Merz (peer reviewer).

The reviewers have discussed the reviews with one another and the Reviewing Editor has drafted this decision to help you prepare a revised submission.

Summary:

The authors previously determined the active, polymerizing conformation of Snf7 as part of ESCRT-III function in MVB formation. This Research Advance exploits these structural data in combination with genetic screens to understand how the active conformation of Snf7 is regulated. They show that the two N-terminal α helices of Snf7 are essential for Vps20 (the most proximal upstream component) regulation, and find mutations that can bypass upstream ESCRT-I and II mutants. Intriguingly, although ESCRT-I and II cluster ubiquitylated cargo, the activated Snf7 mutants still require such cargo for MVB formation. Perhaps the most interesting finding is that Bro1, which is known to bind to Snf7 helix 6, does so to relieve autoinhibition of Snf7, demonstrating that Snf7 helix 6 also confers autoinhibition to Snf7 filament assembly in addition to the previously documented autoinhibition that prevents remodeling of helices 3,4 and 5. This two-stage mechanism of activation is intriguing and probably adds an important level of regulation to the system. Overall, this is a logical follow up to the previous *eLife* paper and adds considerable new data to understanding regulation of ESCRT-III.

The authors should address the following in a revised manuscript:

1) More background is needed for the non-specialist – it is a bit too 'insider'. The Introduction is very minimal, and going through the mutation data on ESCRT-I and -II suppression is difficult without more background on the roles of some of these factors. For example, apart from a cartoon no background is provided regarding the 'arms' of the ESCRT-II complex.

2) In the CD experiments shown in Figure 1, a reference is needed to explain the red shift of the spectrum in the presence of liposomes. In general, more helical structure is associated with increased negative ellipticity at 208 and 222 nm, not a spectral shift. Also, comparing 1F and 1G, why is there no change with WT in 1G when there is 41% binding in 1F? Is this due to use of the truncated construct? This is not made clear.

3) Figure 2; Figure 3—figure supplement 6: The variation in ILV size relies on samples prepared from *vam7^tsf^* vacuole-fusion-deficient spheroplasts fixed in conventional fixative. This accounts for the relatively poor preservation of the MVB morphology. Osmotic gradients during fixation or dehydration probably account for the morphological defects and the larger mean vesicle sizes versus MVBs in the same strains prepared by high-pressure freezing and automated freeze-substitution. The limitations of this approach should be mentioned particularly since there is a strong possibility that the composition (lipids, proteins, lipid:protein ratios) in the vesicles of suppressed cells vary, resulting in divergent robustness to the stresses of conventional fixation. Also, the legend for Figure 2 should specify whether n=150 refers to the number of ILVs, MVBs, or cells, and whether n=150 summed over the experiment or per-treatment. Finally, could some of the size differences be due to the somewhat smaller amount of Snf7 mutant protein relative to WT (Figure 2—figure supplement 2)?

4) At the end of the subsection “Auto-activated Snf7 bypasses Vps20“: an alternative possibility is that the variation in ILV sizes is due to changes in disassembly of the ESCRT III polymers by Vps4. The possibility that the vesicle sizes reflect divergent ESCRT III dynamics (vs. static architecture) is underscored by previous work (Odorizzi et al., 2006; 2010) showing that mutants deficient for Did2, a Vps4 ATPase regulator, have vesicle morphology phenotypes highly similar to those described in the present paper.

5) The MVB sorting assays should be briefly described in Methods. In particular, the authors should reiterate how 0% and 100% are calibrated, and the figure legends need to specify the number of independent experiments summarized for each panel or experiment.

6) A concern in the experiments in Figure 3 is that expression of Hse1-DUB might mildly interfere with normal Hse1 function through a mechanism other than de-ubiquitination. Such a defect might not be evident in a WT background where MVB formation is robust, but it could potentially manifest in the suppressed background where MVB formation might be barely possible. The standard control in experiments using the Piper group's Hse1-DUB fusions is to use fusions containing a catalytically dead DUB domain. These controls need to be included in a supplemental panel. The related experiments in Figure 4 (Vps27 and Bro1 Ub-interaction mutants) are strong, however, so it is recommended that either the Hse1-DUB experiment be omitted, or that it should be presented with these needed controls.

---

## [Author Response]

*The authors should address the following in a revised manuscript: 1) More background is needed for the non-specialist – it is a bit too 'insider'. The Introduction is very minimal, and going through the mutation data on ESCRT-I and -II suppression is difficult without more background on the roles of some of these factors. For example, apart from a cartoon no background is provided regarding the 'arms' of the ESCRT-II complex.*

We agree with the reviewers that more introduction and background information is needed for general readers. We have now expanded the Introduction, and included more background information regarding the key functional and structural features of each ESCRT complex when going through the mutational data on ESCRT-I and ESCRT-II suppression in Figure 3 in the subsection “Auto-activated Snf7 bypasses ESCRT-I and ESCRT-II”.

*2) In the CD experiments shown in Figure 1, a reference is needed to explain the red shift of the spectrum in the presence of liposomes. In general, more helical structure is associated with increased negative ellipticity at 208 and 222 nm, not a spectral shift.*

We have now included two references of CD spectra in the second paragraph of the subsection “Screening for Vps20-independent Snf7 activation mutants”. This seems to be a common observation. We also agree with αthe reviewers that more -helical structure gave *decreased* ratio of molecular ellipticity at negative bands of 208 and 222 nm. We havecorrected this in the aforementioned paragraph.

*Also, comparing 1F and 1G, why is there no change with WT in 1G when there is 41% binding in 1F? Is this due to use of the truncated construct? This is not made clear.*

Yes, the discrepancy is due to the use of a truncated construct. The liposome sedimentation experiment (Figure 1) is performed using intact full-length Snf7 proteins. To observe the major structural rearrangement in the core domain by CD spec (Figure 1), however, we utilized a construct of Snf7 that only contains α1-α4, Snf7^α1-α4^. We have previously reported in Buchkovich et al., Dev Cell2013, Snf7^α1-α4^ (a.k.a. ΔN-Snf7^core^) has a reduced membrane-binding affinity (at ~5%) due to 1) the deletion of the N-terminal amphipathic ANCHR motif that inserts into the membranes, and 2) the partially buried electrostatic membrane binding surface. Therefore, wild-type Snf7 ^α1-α4^ in 1G has no ellipticity change with and without liposomes. However, activation mutations Q90L and N100I trigger Snf7 core domain conformational change and, thus, expose the electrostatic membrane-binding surface, which resulted in CD spectra change in the presence of liposome. We added a sentence in the second paragraph of the subsection “Screening for Vps20-independent Snf7 activation mutants” to clarify this difference.

*3) Figure 2; Figure 3—figure supplement 6: The variation in ILV size relies on samples prepared from vam7^tsf^ vacuole-fusion-deficient spheroplasts fixed in conventional fixative. This accounts for the relatively poor preservation of the MVB morphology. Osmotic gradients during fixation or dehydration probably account for the morphological defects and the larger mean vesicle sizes versus MVBs in the same strains prepared by high-pressure freezing and automated freeze-substitution. The limitations of this approach should be mentioned particularly since there is a strong possibility that the composition (lipids, proteins, lipid:protein ratios) in the vesicles of suppressed cells vary, resulting in divergent robustness to the stresses of conventional fixation.*

We appreciate this comment. However, our chemical fixation protocols have been wildly used in the field and were treated equally and applied to all experimental samples shown in these figures. We observed ILV formation in the suppressors’ condition, indicating that these suppressors have the ability to generate ILVs in the absence of selected ESCRT mutants. We have now added in the last paragraph of the subsection “Auto-activated Snf7 bypasses Vps20 “and in the subsection “Negative Stain Transmission Electron Microscopy” to comment on this potential technical limitation.

*Also, the legend for Figure 2 should specify whether n=150 refers to the number of ILVs, MVBs, or cells, and whether n=150 summed over the experiment or per-treatment.*

N=150 refers to the number of ILVs summed per sample. We have now included this in the legends of Figure 2 and Figure 3—figure supplement 6.

*Finally, could some of the size differences be due to the somewhat smaller amount of Snf7 mutant protein relative to WT (Figure 2—figure supplement 2)?* Yeast cells used for ILV size quantification (Figure 2 and Figure 3—figure supplement 6) have two populations of Snf7 expressed: the *snf7** suppressor (off the plasmid) and the wild-type *SNF7* (off the chromosome). We thus believe that the functional wild-type Snf7 proteins available for ILV biogenesis should be the same between the mutant and the wild-type cells. But in Figure 2—figure supplement 2, in order to identify the expression levels of different Snf7 mutants, these yeast cells used for Western blotting analysis have only one copy of mutant Snf7 (off the plasmid).

*4) At the end of the subsection “Auto-activated Snf7 bypasses Vps20“: an alternative possibility is that the variation in ILV sizes is due to changes in disassembly of the ESCRT III polymers by Vps4. The possibility that the vesicle sizes reflect divergent ESCRT III dynamics (vs. static architecture) is underscored by previous work (Odorizzi et al., 2006; 2010) showing that mutants deficient for Did2, a Vps4 ATPase regulator, have vesicle morphology phenotypes highly similar to those described in the present paper.*We agree with the reviewers that our observed aberrant ILV is similar to the Greg Odorizzi lab’s *did2* mutants. We have shown in Figure 3 that both Vps2 and Vps4 are required for the suppressor’s pathway, indicating that the disassembly of the ESCRT-III polymer is still dependent upon the AAA-ATPase Vps4 machinery. Although the MIM motifs of ESCRT-III for the recruitment of Vps4 are all intact in our system, we cannot completely rule out the kinetics and dynamics of ESCRT-III have subtle differences between the suppressor mutant and the wild-type cells. We thus have added one sentence and a reference at the end of the subsection “Auto-activated Snf7 bypasses Vps20 “to comment on this concern.

*5) The MVB sorting assays should be briefly described in Methods. In particular, the authors should reiterate how 0% and 100% are calibrated, and the figure legends need to specify the number of independent experiments summarized for each panel or experiment.* We have now included this information in Methods (subsection “Flow Cytometry) and in all figure legends.

6) A concern in the experiments in Figure 3 is that expression of Hse1-DUB might mildly interfere with normal Hse1 function through a mechanism other than de-ubiquitination. Such a defect might not be evident in a WT background where MVB formation is robust, but it could potentially manifest in the suppressed background where MVB formation might be barely possible. The standard control in experiments using the Piper group's Hse1-DUB fusions is to use fusions containing a catalytically dead DUB domain. These controls need to be included in a supplemental panel. The related experiments in Figure 4 (Vps27 and Bro1 Ub-interaction mutants) are strong, however, so it is recommended that either the Hse1-DUB experiment be omitted, or that it should be presented with these needed controls.

We removed the Hse1-DUB experiments, because we agree with the reviewers that the related UIM and UBD mutant experiments in Figure 4 are sufficient. However, although we agree that Hse1-DUB might mildly interfere with normal Hse1 function that is not detectable under WT condition, yeast cells used for these Hse1-DUB experiments express their endogenous wild-type Hse1 from the chromosome, which would provide normal ESCRT-0 function. Interestingly, we have found that *hse1* deletion, which has mild defects in MVB sorting in WT condition, abolishes the *snf7** suppression. We thus have reorganized Figure 3. We now included the observation of *hse1* deletion to complete the global analysis of ESCRT mutations and modified the results in the subsection “Auto-activated Snf7 does not bypass Bro1 and ESCRT-0“.